# African American Farmers' Attitudes and Perceptions Towards an Urban Agriculture Certificate Program

Theoneste Nzaramyimana [1,*], Kathryn Orvis [2], Nathan Shoaf [3], Sait Sarr [1] and Tamara Benjamin [4]

1   School of Agriculture and Natural Resources, Kentucky State University, Frankfort, KY 40601, USA; sait.sarr1@kysu.edu
2   Department of Horticulture and Landscape Architecture, Purdue University, West Lafayette, IN 47907, USA; orvis@purdue.edu
3   Purdue University Cooperative Extension Services, Purdue University, West Lafayette, IN 47906, USA; nlshoaf@purdue.edu
4   American Farmland Trust, Washington, DC 20006, USA; tbenjamin@farmland.org
*   Correspondence: theoneste.nzaramyima@kysu.edu

**Abstract:** Farmers' training through experiential learning plays a crucial role in the success of their farming businesses. Aspiring farmers seek crucial skill sets, including financial management, marketing strategies, pricing, budgeting, whole-farm planning, and long-term decision-making. The objective of this study was to assess African American farmers' attitudes and perceptions towards an urban agriculture certificate program launched to equip them with farming skills to help them become more successful in an urban setting. A mixed-method (qualitative and quantitative) research approach was used to evaluate the impact of this certificate program. Pre- and post-survey questionnaires and interviews were administered to assess farmers' prior urban farming knowledge and skills and if there were any changes after the certificate program was launched. Eighteen participants ($n = 18$) who participated and completed the program were surveyed to measure knowledge and skills gained. Results showed that African American farmers participating in the Gary Urban Agriculture Certificate Program significantly ($p < 0.03$) increased their skill sets and knowledge about farming in comparison to their mean scores from pre- and post-certificate program. When participants were asked to summarize the impact of the course, several mentioned that it helped them develop a comprehensive approach to planning, planting, maintaining, and harvesting the produce of their farms. Narratives from interview discussions also support the survey results in which the majority expressed a positive impact of the certificate in helping them learn about the useful farming resources around them and gain skills in running a farm. Key findings support the concept that the urban agriculture certificate program administered by Purdue University impacted participant knowledge and provided a positive attitude towards farming. This study helped us understand the impact of the certificate program. Results provided greater awareness of creating programs to support the development of Gary urban farmers' ability to become more knowledgeable and successful in their farming endeavors.

**Keywords:** African American; certification; farming; networking; (beginning) urban farmer

## 1. Introduction

According to the United Nations in references [1,2], the global population is projected to reach 9–11 billion people by 2050. This will likely increase the number of people living in urban areas and consequently increase the demand for locally produced foods. Fortunately, urban agriculture continues to increase in most U.S. cities, mainly due to a growing demand by consumers for fresh local produce [3,4]. Indiana school districts spent more than $11 million on local food in the year 2011–2012, with 37% of these schools likely to support local farms through their purchase of local produce [5–7]. Other benefits of urban agriculture include improving food access in food-insecure areas, increasing consumption

of fresh fruits and vegetables, boosting youth development opportunities, creating jobs and other career opportunities, and helping people to start small businesses with minimum startup costs [8].

The City of Gary, located in the northwest corner of Indiana, has limited retail food establishments, and those that exist are predominantly fast-food stores [9]. Community members from these areas travel long distances either by bus or car to obtain fresh produce. The lack of healthy food options is believed to be a contributor to mortality and morbidity in the U.S. [10,11]. Other studies demonstrate that poor dietary practices correlate with an increased risk of cardiovascular diseases [11–16]. It has also been found that people who consume high levels of dietary fat, too little fiber, and limited fresh fruit and vegetables have a higher risk of heart disease, stroke, and cancer(s) [17,18]. Ref. [19] showed that, in general, there is a small number of supermarkets in African American neighborhoods compared to white neighborhoods. This leaves community members with limited options where they can go to buy fresh produce, and they often end up consuming ready-made foods.

There are several studies that show that farmers' training plays a crucial role in their farming businesses. For instance, a study conducted by [20] showed that aspiring farmers seek crucial skill sets, including financial management, marketing strategies, pricing, budgeting, whole-farm planning, and long-term decision-making. Reference [21] showed that beginning farmers prefer to learn through experiential and hands-on learning rather than traditional methods such as a classroom-type approach. These two researchers emphasized continuing education through face-to-face outreach programs that provide knowledge and skill sets to the participants. In addition, a study conducted by [22] showed that participants who engaged in farm training viewed farmer networking as a key component of successful educational programs. Participants enjoyed being able to talk to both older and more experienced producers, leaders in agriculture, and peer producers. The findings of this study also showed that participants were eager to learn about tax laws, contract laws, leasing laws, property rights, water rights, and government and environmental regulations [22]. This study further showed participants' interest in several areas, including agriculture business management skills, accounting, and record-keeping skills needed to help manage expenses, incomes, taxes, and decision-making on the farm or ranch. Participants preferred an interactive environment that allowed them to learn from producers and presenters through small-group learning [22]. One more important point mentioned in this study is the nature of the program content. When asked about their intentions to enroll in this event, several mentioned that they aimed to increase efficiency and learn innovative ideas. Other studies also show that participants who enrolled in farm educational programs viewed the content and information that related to their needs increased operational efficiencies and helped solve problems as beneficial [23–26] In terms of knowledge adoption, ref. [27] found that younger farmers were more likely to embrace newer technologies in comparison to older producers. Ref. [28] also found that agriculture educational programs vitalize new opportunities to generate and exchange information and knowledge for sustainable agriculture for young and beginning farmers.

This certificate program was intended to increase the capacity to sustainably grow fresh produce in an urban setting through education and training, networking, and resource assistance while enhancing the competitiveness of urban farmers through increased sales and engagement with local stakeholders. The program's duration was 12 weeks, and it was delivered both in person and online due to the COVID-19 pandemic. The topics covered included the following: site assessment and environmental safety; urban soil health management; direct sowing; transplanting and propagation; growing vegetables; strategies for organic pest control; organic weed management and irrigation; harvest day techniques for quality, efficiency, and food safety; farm finance expense; income and organizational structure; marketing; sales; and farm equipment and maintenance. These topics were adopted from an urban agriculture certificate program that was developed for Indianapolis (Marion County), Indiana, to fit the specific needs of the Gary population. This certificate program is part of a NRCS-SARE Research and Education project (Increasing

the Sustainable Production and Access of Fresh Produce in Urban Areas of NW Indiana) that was designed to support urban farming capacity building in Northwest Indiana, predominantly in Gary.

This current study explored the role an urban agriculture certificate program provided by Purdue University plays in promoting needed skills for Gary farmers to improve farming systems in urban areas and utilize knowledge to improve production efficiencies.

## 2. Materials and Methods

The approaches described for this study are grounded in Bandura's Social Cognitive Theory [29,30], which theorizes that learning occurs in a social context with a dynamic and mutual interaction of the person, environment, and behavior. Connecting concepts of this study grounded in this theory hypothesizes that people learn from one another via observation, imitation, and modeling.

This study started with a quantitative approach by administrating pre- and post-survey questionnaires and then transitioned into a qualitative phase, using interviews and open-ended questions to a select number of participants. This mixed-method study relied on surveys, interviews, and a review of literature to arrive at an understanding of the role of the urban farming certificate project administered by the Purdue Cooperative Extension in increasing skill sets for Gary farmers.

### 2.1. Data Collection

A pre- and post-survey questionnaire was used to measure changes in participants' attitudes, perceptions, and engagement in urban agriculture. The survey was developed by the evaluation and leadership team specifically for the NRCS-SARE Research and Education project referenced above, including individuals with evaluation expertise, urban farming, and beginning farmer programming expertise. Only eighteen participants ($n$ = 18) who participated and completed the program were willing to take the survey to measure knowledge and skills gained. The survey, consisting of 40 Likert scale and open-ended questions with multiple choices, included demographics, knowledge-gain questions specific to the topics taught during the certificate program, questions about urban farming perceptions, urban farming skills, and expectations of learning in the course. Participants were voluntary, self-selected current and aspiring urban farmers in Gary. Delivery of the survey occurred in person in a paper and pencil format for the pre-survey. Due to the COVID-19 pandemic restrictions, delivery of the post-survey was conducted using online Qualtrics (Qualtrics, Provo, UT, USA). Quantitative data were collected using pre- and post-survey questionnaires. Knowledge questions utilized multiple choice format, and attitude and perception questions were ranked on a 1-to-10-point Likert scale with 1 = no confidence and 10 = very confident.

Follow-up in-depth interviews conducted with a sample of four participants willing to share their lived experiences during the participation of the program ($N$ = 4) determined application of knowledge and acquired skills. Interviews started with baseline open-ended questions about the certificate program and then transitioned into open-ended questions about the participants' experiences. An audio recorder and recordings (Zoom-One Platform to Connect) were used to capture responses for all interviews, ensuring accuracy of responses and gestures, and emotional expressions captured additional interesting information and behavior regarding participants' experiences.

### 2.2. Data Analysis

The Statistical Package of the Social Scientist (SPSS®) Version 26 was used to analyze participants' responses across all quantitative items. Descriptive statistics, including means and standard deviations, were used to analyze data. A *t*-test was conducted to compare the mean scores before and after the certificate program.

Qualitative data were collected using audio recordings that were later transcribed by Rev.com company (www.rev.com) and were followed by coding keywords and phrases

from participant interviews for the purpose of developing categories and generating themes that emerged from the interviews. Generated themes provided the framework for the substance analysis of data [31,32]. The process of reading helped build meaning that was chronicled and summarized [33,34]. Coding processes were based on the research objective regarding the perception of participants' attitudes towards urban farming certificates.

## 3. Results and Discussion

The surveys and interviews assessed the participants' attitudes and perceptions towards the urban agriculture certificate. The points of focus included networking opportunities, confidence after the program, clarity of content, knowledge sharing, tool sharing, knowledge gained, and the application for the knowledge because of attending the certificate program.

### 3.1. Quantitative Results

In Table 1, survey results show that African American farmers participating in the Gary Urban Agriculture Certificate Program significantly ($p < 0.03$) increased their skill sets and knowledge about farming in comparison to their mean scores from the pre- and post-certificate program. This aligns with a study conducted by [20] that showed that aspiring farmers seek crucial skill sets, including financial management, marketing strategies, whole-farm planning, and long-term decision-making.

**Table 1.** Changes in the level of clarity, confidence, and understanding of participants of the urban agriculture certificate program.

| Items | Pre (Mean) | Post (Mean) |
|---|---|---|
| Clarity of purpose in developing your urban farm | 7.74, SD = 2.02 | 8.77, SD = 1.58 |
| Confidence in beginning or further developing urban farm project | 5.89, SD = 3.41 | 8.91, SD = 1.72 |
| Level of understanding of the expenses in your farm | 5.16, SD = 2.65 | 7.83, SD = 1.72 |

Overall, there was an increase in mean scores from before and after the urban agriculture certificate program, as displayed in Table 1. This means that the participants had an increase in clarity of purpose in developing their urban farms, confidence, and understanding of the expenses in their farms at the end of the certificate program as compared to prior to the certification process.

A *t*-test was conducted to compare the mean scores of participants ($n = 18$) before and after the urban agriculture certificate program, as shown in Table 2. By comparing the mean scores from before and after the certificate program, there was a statistically significant ($p < 0.03$) difference in confidence, clarity, and understanding of farming for participants before and after the certificate program (t (3) = −3.97, $p = 0.028$). These results suggest that the urban agriculture certificate program influenced participants' clarity of purpose in developing urban farms, confidence in beginning or further developing urban farms, and level of understanding of the expenses of the farm. These results indicate that the participants may perform (more successfully) in their farming businesses because of attending this urban farming certificate program. This aligns with other studies that found that participants who enrolled in farming educational programs viewed the content and information that related to their needs, and this increased their efficiency in operation and helped them solve problems [23–26].

Overall, there was an increase in mean scores of variables from before and after the urban agriculture certificate program (Table 3). This indicates that participants had an increase in networking with other farmers/professionals, gained knowledge about urban farming and the resources around them, shared knowledge and skills, learned how to share tools, and were able to apply the knowledge gained in their farming businesses at the end of the certificate program as compared to prior to the certificate program. This aligns with a previous study conducted by [22] that revealed that participants who engaged in

farm training viewed farmer networking as a key component of successful educational programs, and they preferred an interactive environment that allowed them to network and learn from other stakeholders (including experienced producers, policymakers, and industry experts).

**Table 2.** Paired *t*-test from pre- and post-mean scores of the level of clarity, confidence, and understanding of participants of the urban agriculture certificate program.

| | Paired Differences | | | | | | | |
| | | | 95% Confidence Interval of the Difference | | | | |
| | Mean | Std. Deviation | Std. Error Mean | Lower | Upper | t | df | Sig. (2-Tailed) |
|---|---|---|---|---|---|---|---|---|
| Pre (Mean)–Post (Mean) | −1.99 | 1.00 | 0.50 | −3.58 | −3.97 | −3.97 | 3 | 0.03 |

**Table 3.** Increase in knowledge, accessing new urban farming resources, tool sharing, sharing knowledge, application of knowledge, and networking of the urban agriculture certificate participants.

| Items | Pre (Mean) | Post (Mean) |
|---|---|---|
| Meeting new people because of the workshop (classmates, speakers, instructors, people met while completing the assignments) | 6.89, SD = 1.59 | 8.23, SD = 1.48 |
| Learning about urban farm projects because of this workshop (could include classmates project, guest speakers or others) | 7.47, SD = 1.64 | 9.08, SD = 0.86 |
| Accessing new urban agriculture related resources because of participating in this course | 7.05, SD = 1.87 | 8.67, SD = 1.49 |
| Expanding produce production on my farm | 7.35, SD = 1.61 | 7.31, SD = 2.87 |
| Utilize tool share program to grow my farm | 6.95, SD = 1.54 | 7.33, SD = 2.75 |
| Sharing new things, I learned in this course with other urban farmers or new farmers. | 7.53, SD = 1.74 | 8.33, SD = 1.55 |
| My ability to apply what I learn in this course to my own farm. | 8.05, SD = 1.58 | 9.00, SD = 1.29 |

For the certificate program, a *t*-test was conducted to compare the mean scores of participants before and after the urban agriculture certificate program (*n* = 18) (Table 4). By comparing the mean scores from before and after the certificate program, there were statistically significant ($p < 0.05$) differences for an increase in knowledge, accessing new urban farming resources and tools, knowledge sharing, and application of knowledge for participants before and after the certificate program (t (6) = −3.96, $p = 0.007$). These results suggest that the participants may perform successfully in their farming businesses because of attending this urban farming certificate program.

**Table 4.** Paired *t*-tests from pre- and post-mean scores for increase in knowledge, accessing new urban farming resources, tool sharing, sharing knowledge, application of knowledge, and networking of participants of the urban agriculture certificate.

| | Paired Differences | | | | | | | |
| | | | 95% Confidence Interval of the Difference | | | | |
| | Mean | Std. Deviation | S td. Error Mean | Lower | Upper | t | df | Sig. (2-Tailed) |
|---|---|---|---|---|---|---|---|---|
| Pre (Mean)-Post (Mean) | −0.95 | 0.63 | 0.24 | −1.53 | −0.36 | −3.96 | 6 | 0.007 |

Additional Wilcoxon signed rank test table shown below (Table 5) indicates a significant difference between the pre- and post-tests, with a *p*-value of 0.002 and a *z*-statistic of −3.170, implying a positive impact of the urban farm certificate intervention. These results show that the program brought a significant positive change in knowledge and skill sets among the participants who engaged in this urban farming certificate.

**Table 5.** Wilcoxon signed-rank test for the mean scores from pre- and post-surveys.

| | | | | | | | |
|---|---|---|---|---|---|---|---|
| **Descriptive Statistics** **Test Statistics** [a] | | | | | | | |
| | **N** | **Mean** | **Std. Deviation** | **Minimum** | **Maximum** | | **Post-Post-Pre-Test** |
| Pre-test | 14 | 4.7311 | 0.3824 | 3.9411 | 5.2352 | Z | −3.170 [b] |
| Post-post | 14 | 5.2041 | 0.3028 | 4.5714 | 5.5714 | Asymp. Sig. (2-tailed) | 0.002 |

a. Wilcoxon signed-rank test. b. is based on negative ranks.

### 3.2. Qualitative

African American farmers' responses demonstrated how the certificate program significantly contributed to their networking with other farmers and professionals and increased their knowledge and confidence in their farming activities. Based on the narratives from the experiences lived by the participants in this certificate program, the themes generated were *networking*, *knowledge*, *confidence*, and *value*.

**Networking**—During one-on-one interview discussions, one of the participants mentioned, "Yes, absolutely, I have some people's contact information from the class, and we still do talk and communicate on a regular basis". The tone and expressions made by this participant during the discussion show how much this participant has made network connections with other participants as a result of this program. Another participant said, "So, it is very good to be around a bunch of other people who have the same ideas, not quite the same goals as you maybe, but they bring up questions that you did not even think of. So, it was a good setting to me". Networking opportunities, especially among beginning farmers, play a crucial role in their farming success [22]. Farmers exchange information and explain the challenges they face in their farming businesses. Networking helps them to easily share each and every one of their failures and successes. This is especially important, especially for the participants who often operate in their own isolation. The urban agriculture certificate program provided an intervention to boost the networking among the participants, and this can be seen from both interviews and surveys collected. This aligns with the study carried out by [22] that showed that participants who engaged in farm training viewed farmers networking as a key component of successful educational programs. Participants in this study enjoyed being able to talk to both older or experienced producers, leaders in agriculture, and peer producers. Another theme generated from the interview data was the knowledge gained from the urban agriculture certificate by the participants. During the interview questions, participants expressed how the certificate was informative. Their narratives showed that the certificate contained information that would contribute to their farming successes. The program provided new knowledge and skill sets in farming that they did not have before. Many of these farmers are now engaging together in the internship program to make deeper connections among all of them. They are also creating a networking cohort that will interact and share for years to come.

**Knowledge**—During the interview discussions, one of the participants in this program mentioned: "Soil remediation and marketing, those were very, very, very important for me. Those were some of the subjects we talked about". Another participant also acknowledged: "I know I learned about drainage. I learned about, I think it's called a berm where you, instead of using a raised bed, you just pile on, what is it, mulch and then dirt and things like that. And I learned about how you don't have to spend a lot of money. I was interested

in the hoop house that they showed and how that works. And about the urban farms that are done right here in Chicago, well, right next door in Chicago that are on top of buildings. And you know, that interested me because I feel like that is something that could happen in Gary as well". These two participants' narratives clearly show that this program was useful to them. The certificate program contained knowledge that provided skill sets to the participants who were engaging in the urban farming businesses. This urban farming certificate is one of the few programs that was launched in this community. Based on participants' testimonies, it is evident that the knowledge gained from this program, including soil management, seeding, harvesting, and marketing, will help these farmers successfully grow fresh produce to serve the Gary community and be able to generate income that will contribute towards community development. This finding is supported by [21], which emphasizes that continuing education through face-to-face outreach programs provides knowledge and skill sets to participants.

**Confidence**—Besides the knowledge gained and the networking opportunities, this study also assessed the extent this urban farming certificate increased African American farmers' confidence in reaching out to other farmers. This was shown during the one-on-one interview discussions with those who participated in the urban farming certificate program. One participant mentioned: "*That everybody is kind of standoffish and kind of scared to talk because once you break the ice, people come out their shells and they will start talking. Well, I am around the corner, once you break the ice, it is just even people network and things like that. So, my experiences have been good overall. Overall, I had good experiences*". The gestures and facial expressions displayed by this participant clearly indicated that the urban farming certificate program not only provided knowledge about urban farming but also helped participants come out of their comfort zones. The participant acknowledged how, during this program delivery, participants were scared to open up, but as time went by, they started to share their experiences and connect with each other [23–26].

**Valuable**—This study also assessed the perception of this urban farming certificate by the participants who engaged in this program. The narratives from the participants about the value of the urban farming certificate are highlighted. During the one-on-one interview discussions, one participant mentioned: "*So, it was very, very beneficial to take the class on how to grow and sell. I mean, it is a great opportunity. You learn a lot and you can brush up on skills. It only takes up maybe one or two days out the week, so it is? not time consuming*". The enthusiasm and the excitement of this participant can easily show how the program was so important. The participants showed that the class was beneficial and had lots of skill sets to offer to them [21]. Another participant mentioned: "*It is important to me because, like I said, this is a food dessert. We have a lot of seniors here who don't have access to good and healthy food. I think it was the business portion of the class where they were showing different cities that do urban farming and how it could turn the city around, that interested me because our city does have a green urbanism*". This participant clearly shows how encouraged and grateful they were after attending this urban agriculture certificate program. The participant pointed out the experience she had after being shown what urban farming can do using other cities as examples. The participant's experience and knowledge acquired during the certificate program delivery shows that the program helped her to see what others are doing differently and how one can adopt these same methods to effectively and efficiently produce more and better qualities of fresh food. This is another indication that the program was important for both participants and the community at large.

Another participant also mentioned: "Well, I actually have had this conversation with someone who was interested and what I said was it was very informative if you think? you know a lot about farming, which I did not think I knew anything, but even if you think you know everything it's good to be in this program because you get different insights. And I really enjoyed that actual farmers came into the class and talked to us. We got to talk to people, see the tools. So, you can always learn something". During the interview discussions, the enthusiasm and gestures clearly indicated how appreciative the participants were towards the urban farming certificate program.

## 4. Conclusions

The objective of this study was to assess the participants' perceptions and attitudes towards the certificate program in which they were engaged. The research objective was used to frame the evaluation and explore the contributions of the certificate in promoting networking among farmers, providing knowledge and confidence in running a farm, the clarity of the content, and the importance of the certificate to the participants and the community in general.

Results showed that the certificate program was beneficial to the participants. It helped them to develop some networks with other farmers, even after the program. Participants also mentioned that they gained skill sets and confidence in how to run their farms, sell their produce, and generate income while helping their community grow fresh produce. This was also displayed in the data gathered from the survey questionnaires, where there was an increase in mean scores on the questions asked before and after the certificate program. This aligns with other studies that found that participants who enrolled in farming educational programs viewed the content and information that related to their needs, and this increased their efficiency in operation and helped them solve problems [23–26].

Due to the increasing awareness of the key roles urban farming plays in the sustainability of cities like Gary, Indiana, a collective and more holistic approach is required to raise the benefits of farm certification, especially in underserved communities. This will require the actions of key stakeholders—practitioners, farmers, policymakers, nonprofits, and community leaders—in drafting and evaluating long-lasting and sustainable policy measures and strategies that will enhance urban food security. It also requires investment in training, startup cost programs, or incentives for beginning urban farmers, promoting diversity (including African American Extension personnel), and connecting and sharing resources, ideas, and plans with sister cities, like Chicago, in Illinois.

## 5. Recommendations

The major goal of the certificate program mentioned in this study was to train participants so that they could gain skill sets and develop networks among themselves to increase the supply of fresh produce within the northwest Indiana community of Gary. Due to limited resources such as land and money found in this community, training will not be enough to ensure an increase in fresh produce and income within the community. Therefore, the first recommendation of this study will be for the City of Gary to consider creating some urban farming grants that can be used by those who received training to enable them to start or expand their operations.

Participants acknowledged the issue related to the lack of diversity representation by professionals who trained these farmers during the certificate program. The second recommendation is that organizers should carefully select professionals, especially African American professionals who are representatives of these farmers. This can significantly increase their engagement during their learning activities and break barriers that might hinder them from asking questions or freely interacting with trainers.

The urban farming programs are rapidly increasing because of the increase in number of people who are interested in local produce. This brings future research opportunities for those interested in urban agriculture. As a result, recommendation three is to implement a longitudinal study by incorporating both surveys and interviews for pre- and post-evaluation of future programs. Future interviews should, in addition, include professionals and organizers for these programs to capture the experiences they had with the participants during the program delivery [32].

The fourth and final recommendation is to share with cities within or outside the State of Indiana the content and methods used to evaluate the program to test the reliability and validity of the findings. This will help to improve the methods used or to start using the instrument once it is found to be reliable and valid.

## 6. Implication

This study gives further evidence of the benefits of experiential learning methods with aspiring adult urban farmers. Results demonstrate that the urban agriculture certificate increased participants' knowledge, networking opportunities, and other skill sets necessary to start and sustainably operate an urban farm. Most urban farmers are self-funded, self-taught, and run their urban farming operations in isolation. Therefore, the results from this study imply that the more urban farmers receive training about urban farming, the more they can gain knowledge, skills, and a larger community network. They can also use these resources to produce more fresh produce for the community and, at the same time, generate income from their products.

## 7. Limitations and Assumptions

The major limitation of this study is the small number ($n = 18$) of study participants who completed the certificate program and were able to be surveyed. This was due to the COVID-19 pandemic compelling some participants to drop out of the program due to illness or trying to avoid public gatherings. We assume that the responses of survey participants are truthful and answered to the best of their knowledge and experience.

**Author Contributions:** Conceptualization: T.N. and K.O.; methodology, T.N. and K.O.; software, T.N.; validation, K.O.; formal analysis, T.N.; investigation, T.N and K.O.; resources, T.B.; data curation, T.N.; writing—original draft preparation, T.N.; writing—review and editing, T.N., S.S., K.O., T.B. and N.S.; visualization, T.N.; supervision, K.O.; project administration, T.B. and K.O.; funding acquisition, T.B. All authors have read and agreed to the published version of the manuscript.

**Funding:** Funding for this project was provided by the NRCS-SARE Research and Education project (LNC18-399).

**Institutional Review Board Statement:** This study was conducted according to the guidelines of the Declaration of Helsinki and approved by the Institutional Review Board of Purdue University (protocol originally approved on IRB-2019-137 with modifications approved on IRB-2020-1359). The IRB approval was given before contacting the participants for data collection. All ethical issues were examined by the Purdue University IRB committee, and approval was granted after carefully gathering all the information needed and approaches on how the research was going to be conducted.

**Informed Consent Statement:** Informed consent was obtained from all subjects involved in the study as part of the informed consent process.

**Data Availability Statement:** Data supporting reported results can be found in the attached link: https://projects.sare.org/project-reports/lnc18-399/, (accessed on 5 January 2023).

**Acknowledgments:** The authors extend their gratitude to the survey participants. Additional thanks are given to the City of Gary, Purdue University Extension, and the participants who participated in this study and project.

**Conflicts of Interest:** The authors declare no conflicts of interest related to this research project.

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
