# Peer review of "African American Farmers’ Attitudes and Perceptions Towards an Urban Agriculture Certificate Program"

_urbansci, doi:10.3390/urbansci8040256_

Round 1
Reviewer 1 Report
Comments and Suggestions for Authors
The issue of the current MS is quite interesting and it is significant for the assessment of the importance of agriculture certificate programs.
Below are some suggestions that maybe improve its quality:
The section Abstract should be made more attractive. In this form it is just a stitching of sentences from the text.
Some sentences quoted from the main text constitute details that do not fit into the Abstract. e.g. “At the end of the program, a sample of four participants were conveniently selected from 18 participants and one-on-one interviews were conducted to assess participants’ experiences towards the certificate program”.
In the same section authors write “… was launched (n=18).” The mention of (n=18) should be moved to elsewhere in the sentence, it doesn't make sense in that form.
There are many keywords. 8 keywords are too many. Moreover, some of these are not related to the content, e.g. food security; food production. The word “Gary” is a detail that can be removed.
The section “2.1. Description of Certificate Program” it could be a part of Introduction section and not of Methodology.
In the paper reported that “Eighteen participants who were randomly selected sample willing to participate in the program.”, but it is not clarified the total number of participants in this program. In general, the number of participants is very low and this is the main weakness of the MS. Was the total number of participants in the program small indeed, or only a small number agreed to participate in the current survey? Please explain. Moreover, the above sentence needs to be better expressed.
The statistical analysis is very basic. The application of more reliable statistical tests will enhance the reliability of the results. The McNemar's test is a suitable statistical test for the current survey, to check possible changes “before and after / Pre- and post-survey”. Its application could be a suggestion.
The Discussion section should be improved. Authors selected to unify this section with results, however it will be useful some comparisons with the existence literature to be included.
The conclusion section should be improved. In the current form it is just a summary of the paper. Maybe some suggestions, policy or practical recommendations could be incorporated.
Author Response
Reviewer 1
- “The section Abstract should be made more attractive. In this form it is just a stitching of sentences from the text.”
“The abstract is revised and made more attractive (text changes or additions in red color.”
- “Some sentences quoted from the main text constitute details that do not fit into the Abstract. e.g. “At the end of the program, a sample of four participants were conveniently selected from 18 participants and one-onone interviews were conducted to assess participants’ experiences towards the certificate program.”
“some sentences were removed or adjustments were made where it was necessary.”
- “In the same section authors write “… was launched (n=18).” The mention of (n=18) should be moved to elsewhere in the sentence, it doesn't make sense in that form.”
“(n=18) was moved somewhere with more or better fit information related to the sample size chosen.”
- “There are many keywords. 8 keywords are too many. Moreover, some of these are not related to the content, e.g. food security; food production. The word “Gary” is a detail that can be removed.”
“Some Keywords were removed and the total of the keywords remained were 5.”
- “The section “2.1. Description of Certificate Program” it could be a part of Introduction section and not of Methodology.”
“Section 2.1 was moved to the Introduction section, highlighted in red to fit into the reviewer’s suggestion.”
- “In the paper reported that “Eighteen participants who were randomly selected sample willing to participate in the program.”, but it is not clarified the total number of participants in this program. In general, the number of participants is very low and this is the main weakness of the MS. Was the total number of participants in the program small indeed, or only a small number agreed to participate in the current survey? Please explain. Moreover, the above sentence needs to be better expressed.”
“The issue of sample size was addressed. Only eighteen participants (n=18) who participated and completed the program were willing to take the survey to measure their knowledge and skills gained.” They were not randomly selected, and we corrected that as mentioned before in the text. Also, see “Section 6. Limitations and Assumptions was added to capture the limitations and pitfall for the study.”
- “The statistical analysis is very basic. The application of more reliable statistical tests will enhance the reliability of the results. The McNemar's test is a suitable statistical test for the current survey, to check possible changes “before and after / Pre- and post-survey”. Its application could be a suggestion.”
“The McNemar’s test suggested by the reviewer is a non-parametric test used to test the statistical difference between two groups where the variables are dichotomous in nature. The farmers were not selected based on whether they were farming or not. However, my dataset consists of averages of Likert scale data that does not align with the statistical analysis suggested by the reviewer. I therefore took the liberty of exploring another non-parametric test that better fits my analysis. My research led me to the Wilcoxon Signed Ranks Test, which is another non-parametric test used for paired observations and does not require the dataset to follow a normal distribution.” See the analysis below after performing the Wilcoxon Signed Ranks Test:
|
Descriptive Statistics Test Statistics a |
||||||||
|
|
N |
Mean |
Std. Deviation |
Minimum |
Maximum |
|
Post-post-Pre-test |
|
|
Pre-test |
14 |
4.7311 |
0.3824 |
3.9411 |
5.2352 |
Z |
-3.170b |
|
|
Post-post |
14 |
5.2041 |
0.3028 |
4.5714 |
5.5714 |
Asymp. Sig. (2-tailed) |
.002 |
|
|
a. Wilcoxon Signed Ranks Test |
|
|||||||
|
b. Based on negative ranks. |
|
|||||||
The test statistics table above indicates a significant difference between the pre- and post-tests, with a p-value of 0.002 and a z-statistic of -3.170, implying a positive impact of the Urban Farm Certificate intervention.
- The Discussion section should be improved. Authors selected to unify this section with results, however it will be useful some comparisons with the existence literature to be included.
“The discussion section improved, and the additional text were added and highlighted in red.”
- The conclusion section should be improved. In the current form it is just a summary of the paper. Maybe some suggestions, policy or practical recommendations could be incorporated.
“The conclusion section improved, and additional text is highlighted in red.”
Reviewer 2 Report
Comments and Suggestions for Authors
General Assessment
The submitted article analyzes the attitudes and opinions of African American farmers regarding the urban agriculture certification program led by Purdue University. The study employs a mixed-methods approach, combining pre- and post-training surveys with in-depth interviews to evaluate the program's impact on participants' skills, knowledge, and confidence in urban agriculture. Key findings indicate a significant improvement in participants' goal clarity, knowledge, and ability to develop urban farming initiatives after completing the program.
In my opinion, the article addresses a significant social and environmental issue: access to fresh food in low-income urban areas. The use of mixed methods (surveys and interviews) enhances the credibility of the findings. From a practical perspective, the results can support the development of policies related to urban agriculture and farmer education.
The main weaknesses of the article include the small sample size (n=18 for surveys, n=4 for in-depth interviews), the absence of a control group, and the lack of detailed demographic data about the farmers studied. These limitations restrict the generalizability and interpretability of the findings and make it difficult to attribute the observed effects solely to the program's influence.
Specific Comments
The abstract is concise and contains key information; however, it lacks a clear discussion of the research methods and the significance of the results. It is suggested to clarify the research objectives and include information about the main quantitative and qualitative findings.
The article is written in proper language, but some inconsistencies in terminology are evident (e.g., using "urban agriculture" and "farming/gardening" interchangeably). Ensuring terminological consistency throughout the article would improve clarity.
The methodology section is detailed, but it lacks an explanation for the participant sample selection (e.g., why n=18 was chosen). Additional information on inclusion and exclusion criteria for farmers should be provided. Does the study include all farmers who participated in the training program? Furthermore, the absence of a control group needs justification, as its inclusion could strengthen the reliability of the findings.
The results are presented clearly, but Table 3 contains an error with repeated variables (“Confidence in beginning…”). This needs to be corrected. Additionally, the discussion of research limitations (small sample size, subjectivity in qualitative analysis, etc.) should be expanded. Highlighting these constraints could inspire further research in this area.
Conclusion
In summary, I believe that after addressing the above comments, the article meets the substantive and editorial standards of Urban Science. Therefore, I recommend it for publication.
Author Response
- “The abstract is concise and contains key information; however, it lacks a clear discussion of the research methods and the significance of the results. It is suggested to clarify the research objectives and include information about the main quantitative and qualitative findings.”
“In response to the reviewer suggestions, we addressed it by clearly stating the objectives and added information about the results- both qualitative and quantitative (changes made are highlighted in red on the manuscript.”
- The article is written in proper language, but some inconsistencies in terminology are evident (e.g., using "urban agriculture" and "farming/gardening" interchangeably). Ensuring terminological consistency throughout the article would improve clarity.
“In response to the suggestions from the reviewer, we decided to go with the word “farming” instead of farming or gardening. Changes have been made throughout the document.
- “The methodology section is detailed, but it lacks an explanation for the participant sample selection (e.g., why n=18 was chosen). Additional information on inclusion and exclusion criteria for farmers should be provided. Does the study include all farmers who participated in the training program? Furthermore, the absence of a control group needs justification, as its inclusion could strengthen the reliability of the findings"
“In response to the suggestions from the reviewer, n=18 was chosen because Only eighteen participants (n=18) are the ones who participated and completed the program and were willing to take the survey to measure knowledge and skills gained.” It is also highlighted in the paper under section 2.2.
The absence of a control group was not needed in this case because we want to offer every participant that completed the program equal opportunity to acquire knowledge and skills and to voluntarily take the survey. We can explore the use of control group in the next step of our research. “Also, our IRB was exempted by the Purdue University IRB review Board after carefully checking if there weren’t any ethical issues related to this study. A better
- The results are presented clearly, but Table 3 contains an error with repeated variables (“Confidence in beginning…”). This needs to be corrected. Additionally, the discussion of research limitations (small sample size, subjectivity in qualitative analysis, etc.) should be expanded. Highlighting these constraints could inspire further research in this area.
“Actually, the error with repeated variables was in Table 1 and is corrected. We added a limitation section at the end of the paper. (highlighted in red)”
Round 2
Reviewer 1 Report
Comments and Suggestions for Authors
Authors have adequately addressed most of my concerns/comments in the revised version of the manuscript.